# Closely related type II-C Cas9 orthologs recognize diverse PAMs

Jingjing Wei[1], Linghui Hou[1], Jingtong Liu[1], Ziwen Wang[2], Siqi Gao[1], Tao Qi[1], Song Gao[2], Shuna Sun[3]*, Yongming Wang[1,4]*

[1]State Key Laboratory of Genetic Engineering, School of Life Sciences, Zhongshan Hospital, Fudan University, Shanghai, China; [2]State Key Laboratory of Oncology in South China, Collaborative Innovation Center for Cancer Medicine, Sun Yat-sen University Cancer Center, Guangzhou, China; [3]Children's Hospital of Fudan University, National Children's Medical Center, Shanghai, China; [4]Shanghai Engineering Research Center of Industrial Microorganisms, Shanghai, China

**Abstract** The RNA-guided CRISPR/Cas9 system is a powerful tool for genome editing, but its targeting scope is limited by the protospacer-adjacent motif (PAM). To expand the target scope, it is crucial to develop a CRISPR toolbox capable of recognizing multiple PAMs. Here, using a GFP-activation assay, we tested the activities of 29 type II-C orthologs closely related to Nme1Cas9, 25 of which are active in human cells. These orthologs recognize diverse PAMs with variable length and nucleotide preference, including purine-rich, pyrimidine-rich, and mixed purine and pyrimidine PAMs. We characterized in depth the activity and specificity of Nsp2Cas9. We also generated a chimeric Cas9 nuclease that recognizes a simple $N_4C$ PAM, representing the most relaxed PAM preference for compact Cas9s to date. These Cas9 nucleases significantly enhance our ability to perform allele-specific genome editing.

## Editor's evaluation

This work is relevant to all who are interested in genome editing. The versatile Cas9 nuclease has enabled creative genome editing applications, yet the targetable sequence space is limited by the PAM specificity of the Cas9 RNP. This manuscript expands the Cas9 toolbox by defining the PAM specificity and genome editing activity of a large group of smaller-sized type II-C Cas9s. The results also contribute to our understanding of the diversity of Cas enzymes and show that there is a significant potential in mining for non-trivial genome editing tools amongst highly similar Cas orthologs.

## Introduction

RNA-guided CRISPR/Cas9 was originally identified as part of the microbial adaptive immune system, which contains three crucial components: a Cas9 endonuclease, a CRISPR RNA (crRNA), and a trans-activating CRISPR RNA (tracrRNA) (*Koonin et al., 2017*; *Deltcheva et al., 2011*). These three components constitute an active ribonucleoprotein complex to recognize and cleave foreign DNA (*Gasiunas et al., 2012*; *Jinek et al., 2012*). To recognize foreign DNA, the target sequence should contain (i) a complementary sequence with the crRNA and (ii) a protospacer-adjacent motif (PAM) immediately downstream of the target (*Jinek et al., 2012*). The PAM allows this system to differentiate between the DNA target in invading genetic material (non-self) and the same DNA sequence encoded within CRISPR arrays (self) (*Mojica et al., 2009*).

The competitive coevolution of CRISPR/Cas9 systems with different evolving viruses leads to acceptance of diverse PAMs across the Cas9 nucleases (*Collias and Beisel, 2021*; *Gasiunas et al., 2020*).

*For correspondence:
sun_shuna@fudan.edu.cn (SS);
ymw@fudan.edu.cn (YW)

For instance, SpCas9 recognizes an NGG PAM (*Jiang et al., 2013*; *Qi et al., 2020*; *Wang et al., 2019*; *Xie et al., 2017*); SaCas9 recognizes an NNGRRT PAM (*R*=A, G) (*Ran et al., 2015*); CjCas9 recognizes an N4RYAC (Y=C, T) PAM (*Kim et al., 2017*); St1Cas9 recognizes an NNRGAA PAM (*Agudelo et al., 2020*); BlatCas9 recognizes an $N_4CNAA$ PAM (*Gao et al., 2020*). A recent study screened a list of 79 Cas9 orthologs and identified multiple PAMs, including A-, C-, T-, and G-rich nucleotides, from single-nucleotide recognition to sequence strings longer than four nucleotides (*Gasiunas et al., 2020*).

Several studies have revealed that even phylogenetically closely related Cas9 orthologs may recognize distinct PAMs (*Edraki et al., 2019*; *Hu et al., 2021*). In this study, 'closely related Cas9 orthologs' means that these orthologs have >50% sequence identity and can recognize each other's processed crRNA:tracrRNA duplexes as the guide RNA (gRNA). SpCas9 and SaCas9 are representative type II-A Cas9 nucleases (*Edraki et al., 2019*; *Cong et al., 2013*; *Chatterjee et al., 2018*). ScCas9 has 83.3% sequence identity with SpCas9 and recognizes an NNG PAM (*Chatterjee et al., 2018*). SmacCas9 has 58.4% sequence identity with SpCas9 and recognizes an NAA PAM (*Chatterjee et al., 2020*). We recently identified four Cas9 orthologs that are closely related to SaCas9 and recognized NNGG, NNGRR, and NNGGV PAMs (V=A, C, G), respectively (*Hu et al., 2021*; *Hu et al., 2020*). However, type II-C Cas9 nucleases account for nearly half of the total type II Cas9s (*Shmakov et al., 2017*), but their PAM diversity within closely related orthologs remains largely unknown.

The CRISPR/Cas9 system is a powerful tool for genome editing (*Xie et al., 2017*). By fusing catalytically disabled Cas9 protein to other enzymes, the CRISPR/Cas9 system has been used for base editing (*Gaudelli et al., 2017*; *Komor et al., 2016*), primer editing (*Anzalone et al., 2019*), and gene activation/repression (*Konermann et al., 2015*; *Gilbert et al., 2013*). These new applications need the Cas9-guide RNA (gRNA) complex to precisely target genomic sites. Engineered Cas9 variants with flexible PAMs can increase targeting scope. For example, SaCas9 was engineered to accept an NNNRRT PAM (*Kleinstiver et al., 2015*); SpCas9 was engineered to accept almost all PAMs (*Walton et al., 2020*), but this strategy is time-consuming, and often comes at a cost of the reduced on-target activity. Another strategy is to harness natural Cas9 nucleases for genome editing. We have developed several closely related Cas9 orthologs for genome editing (*Ran et al., 2015*; *Wang et al., 2022*). The advantage of developing tools from closely related Cas9 orthologs is that they can exchange the PAM-interacting (PI) domain. If an ortholog recognizes a particular PAM but does not work efficiently in human cells, we can use this ortholog PI to replace another ortholog PI to generate a chimeric Cas9.

Several type II-C Cas9s, including NmeCas9 (*Hou et al., 2013*), Nme2Cas9 (*Edraki et al., 2019*), CjCas9 (*Kim et al., 2017*), GeoCas9 (*Harrington et al., 2017*), BlatCas9 (*Gao et al., 2020*), and PpCas9 (*Fedorova et al., 2020*) have been developed for genome editing. Nme1Cas9 is a representative type II-C ortholog that was first developed for genome editing in 2013 (*Hou et al., 2013*; *Mali et al., 2013*). It is a compact and high-fidelity enzyme but recognizes a long PAM. To expand the targeting scope, Edraki et al. investigated two Nme1Cas9 orthologs (Nme2Cas9 and Nme3Cas9) and demonstrated that Nme2Cas9 recognized a simple $N_4CC$ PAM (*Edraki et al., 2019*). In this study, we investigated the editing capacity of 29 Nme1Cas9 orthologs, 25 of which were active in human cells. These orthologs recognized diverse PAMs with variable length and nucleotide preference. Importantly, based on these orthologs, we generated a chimeric Cas9 nuclease that recognized a simple $N_4C$ PAM for genome editing. These Cas9 nucleases significantly enhance our ability for precise targeting.

## Results

### Investigation of PAM diversity within Nme1Cas9 orthologs

To investigate the PAM diversity within Nme1Cas9 orthologs, we selected 29 Nme1Cas9 orthologs from the UniProt database (*UniProt Consortium, T, 2018*) using Nme1Cas9 as a reference (*Figure 1—figure supplement 1* and *Table 1*). The amino-acid identities of Nme1Cas9 varied from 59.6% to 70.4%. A previous structural study had shown that residues Q981, H1024, T1027, and N1029 in the Nme1Cas9 PI domain are crucial for PAM recognition (*Sun et al., 2019*). Amino acid sequence alignment revealed that 28 selected orthologs differed in at least one residue corresponding to the four residues of Nme1Cas9 (*Figure 1—figure supplement 2A-C*), indicating that these orthologs may recognize distinct PAMs. BdeCas9 had the same four residues as Nme1Cas9. Nme1Cas9 H1024 forms hydrogen bonds with the fifth nucleotide of the $N_4GATT$ PAM (*Sun et al., 2019*). According to the amino acids corresponding to the Nme1Cas9 H1024, the orthologs selected here could be divided

**Table 1.** Nme1Cas9 orthologs selected from the UniProt database.

| UniProt ID | Host strain | Name | Length (aa) | Identity to Nme1Cas9 (%) |
|---|---|---|---|---|
| A0A011P7F8 | *Mannheimia granulomatis* | MgrCas9 | 1,049 | 65.5 |
| A0A0A2YBT2 | *Gallibacterium anatis IPDH697-78* | GanCas9 | 1,035 | 59.7 |
| A0A0J0YQ19 | *Neisseria arctica* | NarCas9 | 1,070 | 70.4 |
| A0A1T0B6J6 | *[Haemophilus felis]* | HfeCas9 | 1,058 | 65.3 |
| A0A1X3DFB7 | *Neisseria dentiae* | NdeCas9 | 1,074 | 66.4 |
| A0A263HCH5 | *Actinobacillus seminis* | AseCas9 | 1,059 | 66 |
| A0A2M8S290 | *Conservatibacter flavescens* | CflCas9 | 1,063 | 64.2 |
| A0A2U0SK41 | *Pasteurella langaaensis DSM 22999* | PlaCas9 | 1,056 | 63.9 |
| A0A356E7S3 | *Pasteurellaceae bacterium* | PstCas9 | 1,076 | 63 |
| A0A369Z1C7 | *Haemophilus parainfluenzae* | Hpa1Cas9 | 1,056 | 64.8 |
| A0A369Z3K3 | *Haemophilus parainfluenzae* | Hpa2Cas9 | 1,054 | 65.2 |
| A0A377J007 | *Haemophilus pittmaniae* | HpiCas9 | 1,053 | 65.2 |
| A0A378UFN0 | *Bergeriella denitrificans (Neisseria denitrificans)* | BdeCas9 | 1,069 | 68.8 |
| A0A379B6M0 | *Pasteurella mairii* | PmaCas9 | 1,061 | 63.1 |
| A0A379CB86 | *Phocoenobacter uteri* | PutCas9 | 1,059 | 63 |
| A0A380MYP0 | *Suttonella indologenes* | SinCas9 | 1,071 | 67.8 |
| A0A3N3EE71 | *Neisseria animalis* | Nan1Cas9 | 1,074 | 66.6 |
| A0A3S4XT82 | *Neisseria animaloris* | Nan2Cas9 | 1,078 | 65.4 |
| A0A420XER8 | *Otariodibacter oris* | OorCas9 | 1,058 | 64.4 |
| A0A448K7T0 | *Pasteurella aerogenes* | PaeCas9 | 1,056 | 68.2 |
| A0A4S2QB06 | *Rodentibacter pneumotropicus* | RpnCas9 | 1,055 | 63.7 |
| A0A4Y9GBC9 | *Neisseria sp. WF04* | Nsp2Cas9 | 1,067 | 59.6 |
| A6VLA7 | *Actinobacillus succinogenes (strain ATCC 55618/DSM 22257/130Z)* | AsuCas9 | 1,062 | 64.6 |
| C5S1N0 | *Actinobacillus minor NM305* | AmiCas9 | 1,056 | 67.7 |
| E0F2V7 | Actinobacillus pleuropneumoniae serovar 10 str. D13039 | ApsCas9 | 1,054 | 65.4 |
| F2B8K0 | *Neisseria bacilliformis ATCC BAA-1200* | NbaCas9 | 1,077 | 66.5 |
| J4KDT3 | *Haemophilus sputorum* | HspCas9 | 1,052 | 65.2 |
| V9H606 | *Simonsiella muelleri ATCC 29453* | SmuCas9 | 1,063 | 62.2 |
| W0Q6X6 | *Mannheimia sp. USDA-ARS-USMARC-1261* | MspCas9 | 1,047 | 65.7 |

into three groups, which contained aspartate (D), histidine (H), and asparagine (N) residues, respectively (*Figure 1—figure supplement 2A-C*).

Next, we investigated the conservation of the CRISPR direct repeat sequences and tracrRNA sequences among Nme1Cas9 orthologs. Both direct repeats and putative tracrRNAs were identified for 26 Cas9 orthologs. Sequence alignment revealed that direct repeats and the 5' end of tracrRNAs were conserved among Nme1Cas9 orthologs (*Figure 1—figure supplement 3A-B*). We generated

single guide RNA (sgRNA) scaffolds for these orthologs by fusing the 3′ end of a truncated direct repeat with the 5′ end of the corresponding tracrRNA, including the full-length tail, via a 4-nt linker. The RNAfold web server predicted that these sgRNAs contained a conserved stem loop at each end, similar to the Nme1Cas9 sgRNA (*Figure 1—figure supplement 4*). Twenty orthologs contained one stem loop in the middle, while six orthologs contained two stem loops in the middle.

Next, the human-codon-optimized Nme1Cas9 orthologs were synthesized and cloned into the Nme2Cas9_AAV plasmid backbone (*Edraki et al., 2019*). The expression of each Cas9 protein was confirmed by Western blot (*Figure 1—figure supplement 5*). We used a previously developed PAM-screening assay (*Hu et al., 2020*) to determine their PAMs. This is a GFP-activation assay where a 24-nt protospacer followed by an 8 bp random sequence is inserted between the ATG start codon and GFP-coding sequence, resulting in a frameshift mutation. The library was stably integrated into the human genome (HEK293T cells) using a lentivirus. If a Cas9 ortholog enables editing the protospacer, it will generate insertions/deletions (indels) at the protospacer and induce GFP expression in a portion of cells (*Figure 1A–B*). The Nme1Cas9 sgRNA scaffold was used for all Cas9 orthologs in this study. Each Cas9 expression plasmid was co-transfected with an sgRNA plasmid into the cell library. The cells without transfection were included as a negative control, and Nme2Cas9 was included as a positive control. Five days after transfection, GFP-positive cells were observed for 25 orthologs through fluorescence microscope. The percentage of GFP-positive cells varied from ~0.01% to 0.18%, as revealed by flow cytometry analysis (*Figure 1C*).

Next, we analyzed the PAM for each active ortholog. The GFP-positive cells were sorted by flow cytometry, and the protospacer and the 8 bp random sequence were PCR-amplified for deep sequencing. The sequencing results revealed that indels were generated at the target site (*Figure 2A*). We generated the WebLogo diagram for each ortholog based on deep sequencing data. Typically, Nme1Cas9 orthologs displayed minimal or no base preference at PAM positions 1–4. For the orthologs that potentially require PAMs longer than 8 bp, we shifted the target sequence by three nucleotides in the 5′ direction to allow PAM identification to be extended from 8 to 11 bp.

Interestingly, Nme1Cas9 orthologs recognized diverse PAMs (*Figure 2B–C* and *Figure 2—figure supplement 1*). Generally, they displayed strong nucleotide preferences for up to four base pairs. When orthologs contained an aspartate (D) residue corresponding to the Nme1Cas9 H1024, they displayed a strong C preference at PAM position 5 (*Figure 2B*); when orthologs contained histidine (H), or asparagine (N) residues corresponding to the Nme1Cas9 H1024, they displayed a strong R ($R=A$ or G) preference at PAM position 5 (*Figure 2C*). The length of PAM recognition varied between 5 and 11 base pairs. Nme2Cas9 recognized an $N_4CC$ PAM, consistent with a previous report (*Edraki et al., 2019*). BdeCas9 recognized an $N_4GATT$ PAM that was the same as Nme1Cas9 (*Hou et al., 2013*). In addition, many orthologs displayed degenerate PAM recognition (e.g. HpiCas9, MgrCas9, and NbaCas9).

We have generated calculated structural models of these orthologs in complex with sgRNA and DNA using the crystal structure of Nme1Cas9 (PDB ID: 6JDV). Some specificity shifts can be well explained by these structural models. When the amino acid near the 5 position of the PAM is histidine, its side chain forms a potential hydrogen bond with the 6-hydroxyl group of guanine. Replacement of this guanine by cytosine or thymine would cause a major clash, whereas adenine lacks the hydroxyl group to form hydrogen bond with the histidine (*Figure 2—figure supplement 2A*). Likewise, an aspartate at 5 position of the PAM would favor a specific recognition of cytosine via hydrogen bonding with its 4-amine group, but not of other bases that may either result in major clash or abolish the hydrogen bond (*Figure 2—figure supplement 2B*). Similar explanation applies also to the apparent specificity between glutamine and adenine at the 8 position of the PAM on the target sequence (TS, *Figure 2—figure supplement 2C*). Since Nsp2Cas9 induced more GFP-positive cells (0.18%) than other Cas9s and recognized a simple $N_4CC$ PAM, we focused on Nsp2Cas9 in the following study.

## Nsp2Cas9 enables genome editing for endogenous sites

We tested the optimized crRNA length for Nsp2Cas9. We designed a series of crRNAs varied from 18 to 26 nt to target *GRIN2B* gene. The results showed that 22–26 nt crRNA activities were comparable (*Figure 3—figure supplement 1*). The 22 nt crRNA was used in the following study. To test whether Nsp2Cas9 enables genome editing for endogenous targets in human cells, we selected a panel of 19 targets. Nme2Cas9 has the same expression backbone as Nsp2Cas9 and was used for side-by-side

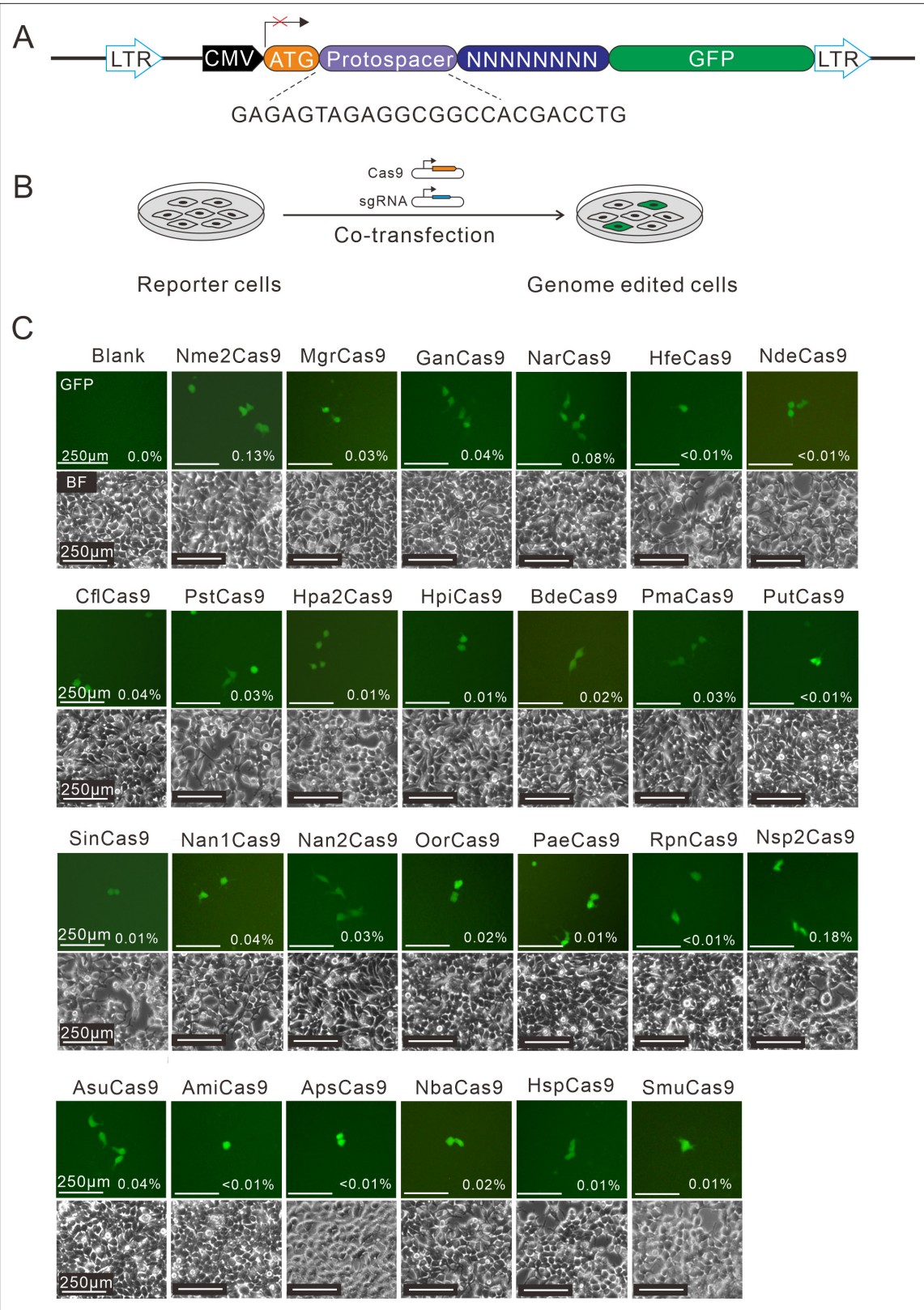

**Figure 1.** Screening of Nme1Cas9 orthologs activities through a GFP- activation assay. (**A**) Schematic of the GFP-activation assay. A protospacer flanked by an 8 bp random sequence is inserted between the ATG start codon and GFP-coding sequence, resulting in a frameshift mutation. The library DNA is stably integrated into HEK293T cells via lentivirus infection. Genome editing can lead to in-frame mutation. The protospacer sequence is shown below. (**B**) The procedure of the GFP-activation assay. Cas9 and sgRNA expression plasmids were co-transfected into the reporter cells. GFP-positive cells could

*Figure 1 continued on next page*

*Figure 1 continued*

be observed if the protospacer is edited. (**C**) Twenty-five out of 29 Nme1Cas9 orthologs could induce GFP expression. The percentage of GFP-positive cells is shown. Reporter cells without Cas9 transfection are used as a negative control. Scale bar: 250 μm.

The online version of this article includes the following source data and figure supplement(s) for figure 1:

**Source data 1.** The maps of all plasmids used in the study for *Figure 1*.

**Figure supplement 1.** Phylogenetic tree of the selected Nme1Cas9 orthologs.

**Figure supplement 2.** Alignment of the PI domain of Nme1Cas9 orthologs.

**Figure supplement 3.** The alignment of direct repeats and tracrRNAs of Nme1Cas9 orthologs.

**Figure supplement 4.** Single-guide RNA (sgRNA) scaffolds of Nme1Cas9 orthologs.

**Figure supplement 5.** Protein expression levels of Nme2Cas9 orthologs were analyzed by western blot.

**Figure supplement 5—source data 1.** Source data for *Figure 1—figure supplement 5*.

comparison (*Figure 3A*). Western blot analysis revealed that their protein expression levels were comparable (*Figure 3B*). Five days after transfection of Cas9 and sgRNA expression plasmids, cells were harvested, and genomic DNA was extracted for targeted deep sequencing. The results revealed that Nsp2Cas9 and Nme2Cas9 displayed comparable editing activity, although the activities varied depending on the targets (*Figure 3C–D*). We tested Nsp2Cas9 editing capacity in additional cell lines, including HeLa, HCT116, A375, SH-SY5Y, and mouse N2a cells, and it could also generate indels in these cells with varying efficiencies (*Figure 3—figure supplement 2A-E*). In summary, these results demonstrated that Nsp2Cas9 could serve as a new player for genome editing.

It has been reported that residues S593 and W596 in Nme1Cas9 play vital roles in cleavage activity. Replacement of the single or double residues with arginine could increase the cleavage activity of Nme1Cas9 in vitro (*Sun et al., 2019*). To test whether these two residues could influence Nsp2Cas9 activity, we identified the corresponding residues S597 and W600 in Nsp2Cas9 by protein sequence alignment and replaced the single or double residues with arginine. To test the activities of the resulting Cas9 variants, we constructed a GFP-activation cell line that was similar to the PAM-screening construct but with a fixed PAM (*Figure 3—figure supplement 3A*). The editing activities could be reflected by the GFP-positive cells. However, five days after genome editing, the results showed that single or double mutation variants could not improve the editing activities (*Figure 3—figure supplement 3B*).

## NarCas9 enables genome editing for endogenous sites

In addition to Nsp2Cas9, we also tested the editing ability of NarCas9, which recognizes a simple $N_4C$ PAM. Five days after transfection of NarCas9 and sgRNA expression plasmids, the cells were harvested, and genomic DNA was extracted for targeted deep sequencing. The results showed that NarCas9 could generate indels in both HEK293T and HeLa cells, but the efficiency was low (*Figure 3—figure supplement 3A-C*). SmuCas9 also recognizes a simple $N_4C$ PAM, but it exhibited minimal activity in the PAM-screening assay, and we did not study this further.

## A chimeric Cas9 nuclease enables genome editing for endogenous sites

We and others have demonstrated that swapping the PI domain between closely related orthologs can generate a chimeric nuclease that recognizes distinct PAMs (*Edraki et al., 2019*; *Chatterjee et al., 2020*; *Hu et al., 2020*). To expand the targeting scope, we replaced the Nsp2Cas9 PI with SmuCas9 PI, resulting in a chimeric Cas9 nuclease that we named 'Nsp2-SmuCas9' (*Figure 4A*). The PAM-screening assay revealed that Nsp2-SmuCas9 could induce GFP expression (*Figure 4—figure supplement 1A*). Deep sequencing analysis revealed that Nsp2-SmuCas9 recognized an $N_4C$ PAM (*Figure 4B–C*). We tested the activities of Nsp2-SmuCas9 in a panel of 12 endogenous targets that have been used for NarCas9. Targeted deep sequencing revealed that Nsp2-SmuCas9 could efficiently induce indels at these sites (*Figure 4D*). Importantly, Nsp2-SmuCas9 displayed higher activities than NarCas9 (*Figure 4E*).

We also replaced the Nsp2Cas9 PI with NarCas9 PI to generate a chimeric Nsp2-NarCas9; replaced the Nme2Cas9 PI with SmuCas9 PI to generate a chimeric Nme2-SmuCas9; replaced Nme2Cas9 PI with NarCas9 PI to generate a chimeric Nme2-NarCas9. The PAM-screening assay revealed that only

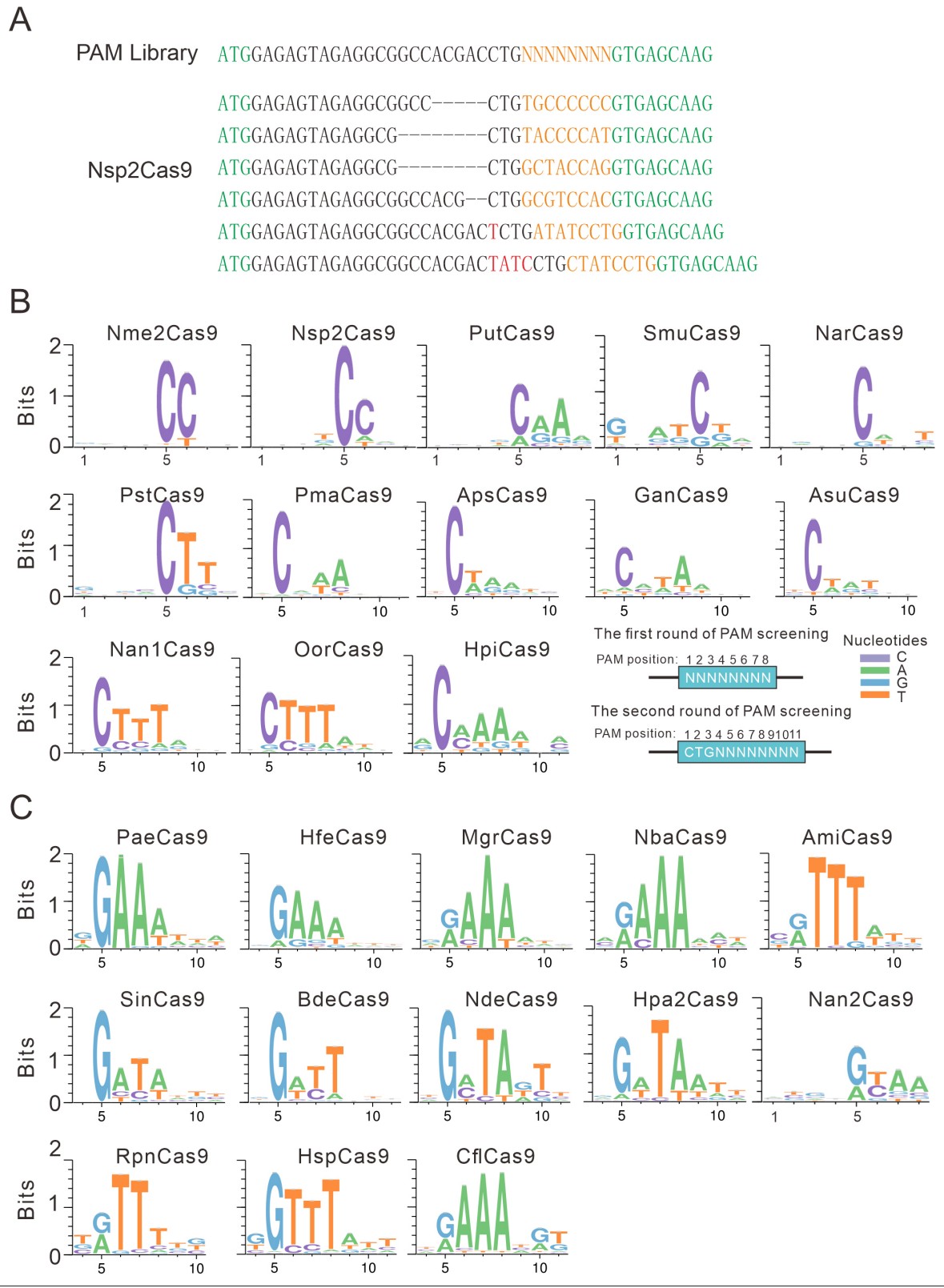

**Figure 2.** PAM analysis for each Cas9 nuclease. (**A**) Example of indel sequences measured by deep sequencing for Nsp2Cas9. The GFP coding sequences are shown in green; an 8 bp random sequence is shown in orange; black dashes indicate deleted bases; red bases indicate insertion mutations. (**B**) The PAM WebLogos for Nme1Cas9 orthologs containing an aspartate residue corresponding to the Nme1Cas9 H1024. PAM positions for each WebLogo are shown below. The PAM WebLogos for Nme2Cas9, Nsp2Cas9, PutCas9, SmuCas9, NarCas9, PstCas9 are generated from the first

*Figure 2 continued on next page*

*Figure 2 continued*

round of PAM screening and the PAM WebLogos for others are generated from the second round of PAM screening. PAM positions in the screening assay are shown on the bottom right. (**C**) The PAM WebLogos for Nme1Cas9 orthologs containing histidine, or asparagine residues corresponding to the Nme1Cas9 H1024. PAM positions for each WebLogo are shown below. The PAM WebLogo for Nan2Cas9 is generated from the first round of PAM screening and the PAM WebLogos for others are generated from the second round of PAM screening.

The online version of this article includes the following source data and figure supplement(s) for figure 2:

**Source data 1.** The number of unique PAM sequences and the median coverage of every individual PAM variant for the *Figure 2B and C*.

**Figure supplement 1.** PAM wheels for Nme1Cas9 orthologs.

**Figure supplement 2.** The specificity between amino acids and bases in calculated structural models.

Nme2-SmuCas9 could induce GFP expression (*Figure 4—figure supplement 1A*). Deep sequencing analysis revealed that Nme2-SmuCas9 recognized an $N_4C$ PAM (*Figure 4—figure supplement 1B-C*). We tested the activities of Nme2-SmuCas9 in a panel of 12 endogenous targets that have been used for NarCas9. Targeted deep sequencing revealed that Nme2-SmuCas9 could induce indels at two sites (*Figure 4—figure supplement 1D*). These data demonstrated that Nme2-SmuCas9 activity is low.

We generated structural models of Nsp2-NarCas9, Nsp2-SmuCas9, and NarCas9 using the crystal structure of highly homologous Nme1Cas9 in complex with sgRNA and dsDNA (PDB ID: 6JDV) as the template by SWISS-MODEL. By superimposing these models, we noticed that residues G1035, K1037 and T1038 of Nsp2-NarCas9 chimera protrude towards the DNA molecule, which would prevent the binding with DNA and thereby abolishing the editing activity (*Figure 4—figure supplement 2A*). In comparison, Nsp2-SmuCas9 and NarCas9, which possess the Cas activity, show no protrusion at the corresponding position (*Figure 4—figure supplement 2B-C*). When Nme2-NarCas9 was analyzed by the same method, 1052 Arg will crash with the DNA strand, leading to a failure of binding with DNA. The predicted result revealed that the Nme2-SmuCas9 1052 Arg also crash with the DNA strand, which may lead to low editing efficiency (*Figure 4—figure supplement 3A-B*).

## Activity comparison

Next, we compared the activity of Nsp2Cas9 and Nsp2-SmuCas9 to the extensively used Cas9 variants, including wild-type SpCas9 (SpCas9-WT) (*Cong et al., 2013*), SpCas9-NG (*Nishimasu et al., 2018*), and SpCas9-RY (*Walton et al., 2020*). We cloned these Cas9 variants into identical plasmid backbones (*Figure 5A*). Western blot analysis revealed that their protein expression levels were comparable (*Figure 5B*). We compared their activities with a panel of 11 targets. After 5 days of genome editing, targeted deep sequencing results revealed that SpCas9 was the most active enzyme. Nsp2Cas9, SpCas9-NG, and SpCas9-RY displayed similar activity. Nsp2-SmuCas9 displayed lower activities than other Cas9 variants (*Figure 5C-D*).

## Analysis of Nsp2Cas9 specificity

Next, we analyzed Nsp2Cas9 specificity by using the GFP-activation assay, and Nme2Cas9 was used for comparison. We generated a panel of 11 sgRNAs with dinucleotide mismatches. Five days after co-transfection of Nsp2Cas9 with individual sgRNAs, GFP-positive cells were analyzed by using a flow cytometer. The results showed that Nsp2Cas9 displayed moderate off-target effects with some sgRNAs (sgRNAs M1, M4, M8, M9, and M10) (*Figure 6A*). In contrast, Nme2Cas9 was a highly specific enzyme that did not tolerate dinucleotide mismatches (sgRNAs M2-M11), consistent with a previous study (*Edraki et al., 2019*).

To further compare the specificity of Nsp2Cas9 and Nme2Cas9, we performed genome-wide unbiased identification of double-stranded breaks enabled by sequencing (GUIDE-seq) (*Kleinstiver et al., 2015*). Three endogenous sites targeting GRIN2B, VEGFA and AAVS1 were selected. Five days after electroporation of the Cas9 expression plasmid with GUIDE-seq oligos into HEK293T cells, libraries were prepared for deep sequencing. The sequencing analysis revealed that both Cas9 nucleases could robustly generate indels at three on-targets, as reflected by the high GUIDE-seq read counts (*Figure 6B*). We only detected one off-target site for Nsp2Cas9 on the GRIN2B target (*Figure 6B*).

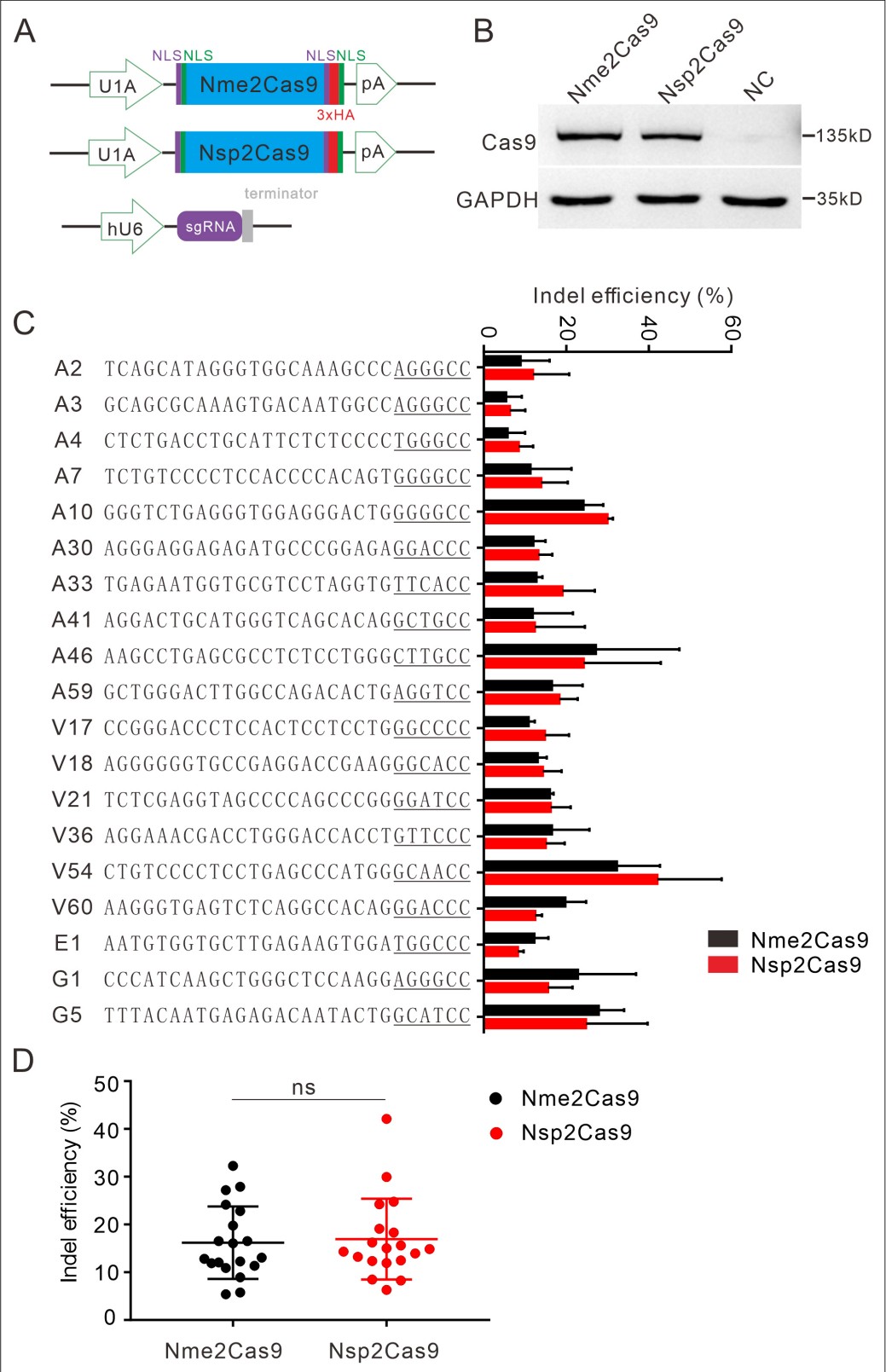

**Figure 3.** Nsp2Cas9 enables editing in HEK293T cells. (**A**) Schematic of Cas9 and sgRNA expression constructs. U1A: U1A promoter; pA: polyA; NLS: nuclear localization signal; HA: HA tag. (**B**) Protein expression levels of Nsp2Cas9 and Nme2Cas9 were analyzed by Western blot. HEK293T cells without Cas9 transfection were used as negative control (NC). (**C**) Comparison of Nsp2Cas9 and Nme2Cas9 editing efficiencies at 19 endogenous loci in

*Figure 3 continued on next page*

*Figure 3 continued*

HEK293T cells. Data represent mean ± SD for n=3 biologically independent experiments. (**D**) Quantification of the indel efficiencies for Nsp2Cas9 and Nme2Cas9. Each dot represents an average efficiency for an individual locus. Data represent mean ± SD for n=3 biologically independent experiments. *P* values were determined using a two-sided Student's *t* test. *P*=0.7486 (*P*>0.05), ns stands for not significant.

The online version of this article includes the following source data and figure supplement(s) for figure 3:

**Source data 1.** Source data for *Figure 3C and D*.

**Source data 2.** Source data for *Figure 2B*.

**Figure supplement 1.** The effect of spacer length on the efficiency of Nsp2Cas9 editing.

**Figure supplement 1—source data 1.** Source data for *Figure 3—figure supplement 1*.

**Figure supplement 2.** Nsp2Cas9 enables editing in different mammalian cells.

**Figure supplement 2—source data 1.** Source data for *Figure 3—figure supplement 2A-D*.

**Figure supplement 3.** Rational engineering of Nsp2Cas9.

**Figure supplement 3—source data 1.** Source data for *Figure 3—figure supplement 3B*.

**Figure supplement 4.** NarCas9 enables genome editing in mammalian cells.

**Figure supplement 4—source data 1.** Source data for *Figure 3—figure supplement 4B*.

**Figure supplement 4—source data 2.** Source data for *Figure 3—figure supplement 4C*.

## Analysis of Nsp2-SmuCas9 specificity

Next, we analyzed Nsp2-SmuCas9 specificity by using the GFP-activation assay. Nsp2-SmuCas9 enzyme showed a minimal or background level of off-target effects with mismatched sgRNAs (*Figure 7A*). Next, GUIDE-seq was performed to test the specificity of Nsp2-SmuCas9. We selected three target sites containing $N_4C$ PAMs. The results revealed that one off-target cleavage occurred on the EMX1 target (*Figure 7B*). Altogether, these results demonstrated that Nsp2Cas9 and Nsp2-SmuCas9 offered new platforms for genome editing.

## Discussion

Type II CRISPR/Cas9, including subtypes II-A, -B, and -C, is the most common class 2 system found in bacteria and archaea (*Makarova et al., 2015*). We and others previously demonstrated that closely related type II-A orthologs preferred variable but purine-rich PAMs (*Hu et al., 2021*; *Chatterjee et al., 2018*; *Chatterjee et al., 2020*; *Hu et al., 2020*; *Wang et al., 2022*). The PAM diversity was also observed within the closely related type V-A orthologs, where Cas12a nucleases recognized the canonical TTTV (V=A/C/G) motif but notable deviations of nucleotide preference existed (*Jacobsen et al., 2020*). In this study, we investigated the PAM diversity within closely related type II-C orthologs. We identified PAMs for 25 Nme1Cas9 orthologs, significantly extending the list of type II-C Cas9 PAMs. These PAMs included purine-rich (e.g. PaeCas9 and HfeCas9), pyrimidine-rich (e.g. Nsp2Cas9 and Nan1Cas9), and mixed purine and pyrimidine (e.g. HpiCas9 and NdeCas9) PAMs, which are more diverse than those identified from type II-A and type V-A orthologs. The PAM length also varied dramatically from a single nucleotide (e.g. NarCas9) to up to 11 nucleotides (e.g. CflCas9). These findings have increased knowledge of PAM diversity within type II-C Cas9s.

Our study offers new Cas9 tools for genome editing. We demonstrated that Nsp2Cas9 was an efficient enzyme for genome editing. By swapping PIs between two Cas9s, we generated a chimeric Nsp2-SmuCas9 that recognizes a simple $N_4C$ PAM. To our knowledge, the $N_4C$ PAM is the most relaxed PAM recognized by compact Cas9 orthologs identified to date. Based on PAMs identified in this study, more chimeric Cas9s can be developed to expand targeting scope. For clinical applications, it is crucial to develop a CRISPR toolbox capable of recognizing multiple PAMs. For example, allele-specific gene disruption through non-homologous end joining (NHEJ) is a potential strategy to treat autosomal dominant diseases (*Gao et al., 2018*; *Diakatou et al., 2021*), where the causative gene is haplosufficient. Autosomal dominant diseases are mainly caused by single-nucleotide missense mutations (*Guo et al., 2013*). If missense mutations form novel PAMs, CRISPR/Cas9 nucleases can disrupt mutant alleles by a PAM-specific approach (*Diakatou et al., 2021*; *Courtney et al., 2016*). With

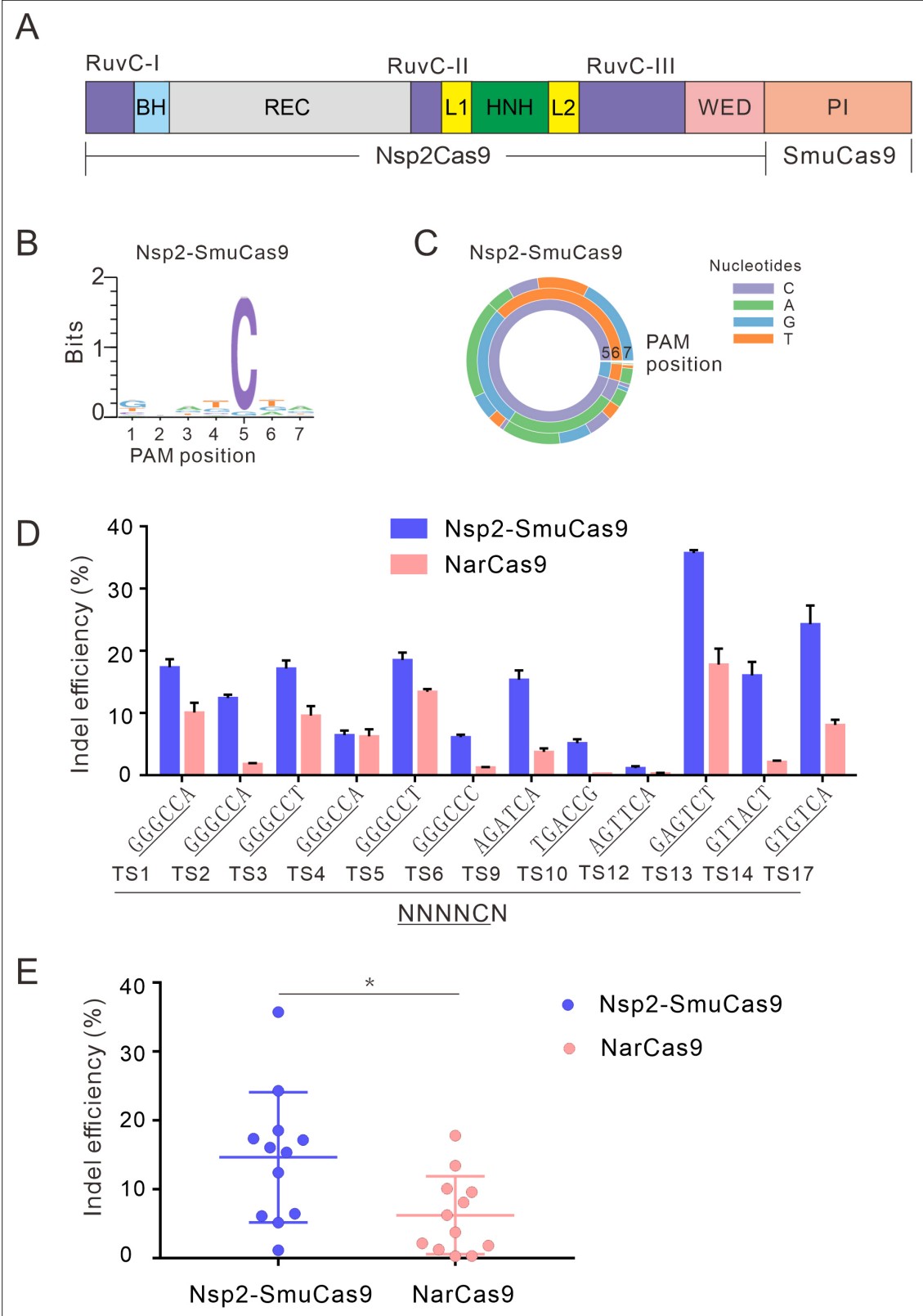

**Figure 4.** Characterization of Nsp2-SmuCas9 for genome editing. (**A**) Schematic diagram of chimeric Cas9 nucleases based on Nsp2Cas9. PI domain of Nsp2Cas9 was replaced with the PI domain of SmuCas9. (**B**) Sequence logos and (**C**) PAM wheel diagrams indicate that Nsp2-SmuCas9 recognizes an $N_4C$ PAM. (**D**) Nsp2-SmuCas9 generated indels at endogenous sites with $N_4C$ PAMs in HEK293T cells. Indel efficiencies were determined by targeted deep sequencing. NarCas9 is used as a control. Data represent mean ± SD for n=3 biologically independent experiments. (**E**) Quantification of the

*Figure 4 continued on next page*

*Figure 4 continued*

indel efficiencies for Nsp2-SmuCas9 and NarCas9. Each dot represents an average efficiency for an individual locus. Data represent mean ± SD for n=3 biologically independent experiments. p values were determined using a two-sided Student's *t* test. *p=0.0148 (0.01<p < 0.05).

The online version of this article includes the following source data and figure supplement(s) for figure 4:

**Source data 1.** Source data for *Figure 4D and E*.

**Figure supplement 1.** Test of 4 chimeric Cas9 activity through a GFP-activation assay.

**Figure supplement 1—source data 1.** Source data for *Figure 4—figure supplement 1D*.

**Figure supplement 2.** Structure of the fully complementary Cas9-sgRNA-dsDNA complex in a catalytic state.

**Figure supplement 3.** Calculated structural models of Nme2-NarCas9 and Nme2-SmuCas9 chimeras.

further development, we anticipate that CRISPR tools from type II-C Cas9 orthologs can play a vital role in therapeutic applications.

# Materials and methods

## Key resources table

| Reagent type (species) or resource | Designation | Source or reference | Identifiers | Additional information |
|---|---|---|---|---|
| *Gene* (*Homo sapiens*) | AAVS1 | GenBank | HGNC:HGNC:22 | |
| *Gene* (*Homo sapiens*) | VEGFA | GenBank | HGNC:HGNC:12680 | |
| *Gene* (*Homo sapiens*) | EMX1 | GenBank | HGNC:HGNC:3340 | |
| *Gene* (*Homo sapiens*) | GRIN2B | GenBank | HGNC:HGNC:4586 | |
| Recombinant DNA reagent | Instant Sticky-end Ligase Master Mix | NEB | Catalog #: M0370S | |
| Recombinant DNA reagent | T4 DNA ligase | NEB | Catalog #: M0202S | |
| Cell line (*Homo-sapiens*) | HEK293T (normal, Adult) | ATCC | CRL-3216 | |
| Cell line (*Homo-sapiens*) | HeLa | ATCC | CRM-CCL-2 | |
| Cell line (*Homo-sapiens*) | SH-SY5Y | ATCC | CRL-2266 | |
| Cell line (*Homo-sapiens*) | A375 (normal, Adult) | ATCC | CRL-1619 | |
| Cell line (*Homo-sapiens*) | HCT116 (adult male) | ATCC | CCL-247 | |
| Cell line (*Mus musculus*) | N2a cells (mouse neuroblasts) | ATCC | CCL-131 | |
| Antibody | Anti-HA (rabbit polyclonal) | abcom | abcam: ab137838; RRID:AB_262051 | (1:1000) |
| Antibody | Anti-GAPDH (rabbit polyclonal) | Cell Signaling | Cell Signaling:#3683; RRID:AB_307275 | (1:1000) |
| Sequence-based reagent | dsODN-F | This paper | dsODN oligo primer | 5'- P-G*T*TTAATTGAGTTGTCATATGTTAATAACGGT*A*T -3' |
| Sequence-based reagent | dsODN-R | This paper | dsODN oligo primer | 5'- P-A*T*ACCGTTATTAACATATGACAACTCAATTAA*A*C –3' |

*Continued on next page*

*Continued*

| Reagent type (species) or resource | Designation | Source or reference | Identifiers | Additional information |
|---|---|---|---|---|
| Sequence-based reagent | P5_index_F | This paper | PCR primers | AATGATACGGCGACCACCGAGATCTACACTGAACCTTACACTCTTTCCCTACACGAC |
| Sequence-based reagent | Nuclease_off_+_GSP | This paper | PCR primers | GGATCTCGACGCTCTCCCTATACCGTTATTAACATATGACA |
| Sequence-based reagent | Nuclease_off_-_GSP1 | This paper | PCR primers | GGATCTCGACGCTCTCCCTGTTTAATTGAGTTGTCATATGTTAATAAC |
| Sequence-based reagent | P5_2 | This paper | PCR primers | AATGATACGGCGACCACCGAGATCTACAC |
| Sequence-based reagent | Nuclease_off_+_GSP2 | This paper | PCR primers | CAAGCAGAAGACGGCATACGAGATTCGCCTTAGTGACTGGAGTTCAGACGTGTGCTCTTCCGATCTACATATGACAACTCAATTAAAC |
| Sequence-based reagent | Nuclease_off_-_GSP2 | This paper | PCR primers | CAAGCAGAAGACGGCATACGAGATCTAGTACGGTGACTGGAGTCCTCTCTATGGGCAGTCGGTGATTTGAGTTGTCATATGTTAATAACGGTA |
| Software, algorithm | FlowJo software | FlowJo VX | | |
| Software, algorithm | CorelDRAW 2020 software | CorelDRAW 2020 | | |
| Software, algorithm | Vector NTI software | Vector NTI | | |
| Software, algorithm | GraphPad Prism software | GraphPad Prism 7 (https://graphpad.com) | RRID:SCR_015807 | Version 7.0.0 |
| Chemical compound, drug | SYBR Gold nucleic acid stain | Thermo Fisher Scientific | Thermo Fisher Scientific: S11494 | |

## Plasmid construction

### Cas9 expression plasmid construction

The plasmid Nme2Cas9_AAV (Addgene #119924) was amplified by the primers Nme2Cas9-F/ Nme2Cas9-R to obtain the Nme2Cas9_AAV backbone. The human codon-optimized Cas9 gene (*Supplementary file 1*) was synthesized by HuaGene (Shanghai, China) and cloned into the Nme2Cas9_AAV backbone by the NEBuilder assembly tool (NEB) according to the manufacturer's instructions. Sequences of each Cas9 were confirmed by Sanger sequencing (GENEWIZ, Suzhou, China). The maps of all plasmids used in the study are packaged in *Figure 1—source data 1*. The primer sequences for the Nme2Cas9_AAV backbone, variants of Nsp2Cas9 and chimeric Cas9 nucleases are listed in *Supplementary file 1*. Single-guide RNA sequences for each Cas9 are listed in *Supplementary file 2*.

### sgRNA expression plasmid construction

The sgRNA expression plasmids were constructed by ligating sgRNA into the Bsa1-digested U6-Nme2_scaffold plasmid, which is the same as Nme1-scaffold. The primer sequences and target sequences are listed in *Supplementary file 3* and *Supplementary file 4*, respectively.

### Cell culture and transfection

The cell culture reagents were purchased from Gibco unless otherwise indicated. HEK293T, HeLa, SH-SY5Y, A375 and N2a cell lines were maintained in Dulbecco's modified Eagle's medium (DMEM). HCT116 cells were cultured in McCoy's 5 A medium (Gibco). All cell cultures were supplemented with 10% fetal bovine serum (FBS) (Gibco) that was inactivated at 56 °C for 30 min and 1% penicillin-streptomycin (Gibco). All cell were cultured in a humidified incubator at 37 °C and 5% $CO_2$. All cell lines identities were validated by STR profiling (ATCC) and repeatedly tested for mycoplasma by PCR.

HEK293T, HeLa, and N2a cells were transfected with Lipofectamine 2000 (Life Technologies) according to the manufacturer's instructions. SH-SY5Y, A375, and HCT116 cells were transfected with

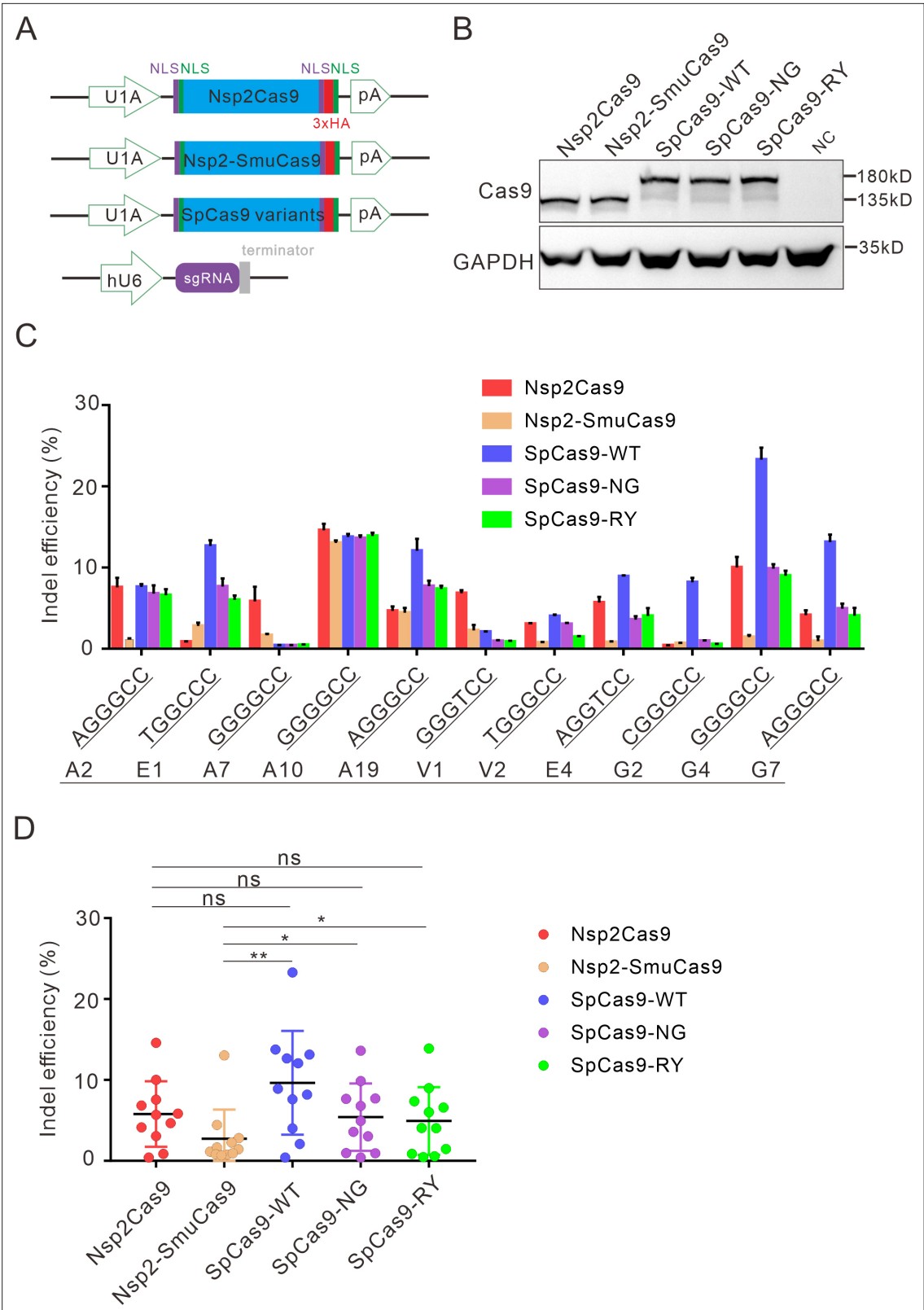

**Figure 5.** Comparison of indel efficiency between Nsp2Cas9, Nsp2-SmuCas9, and SpCas9. (**A**) Schematic of Cas9 and sgRNA expression constructs. U1A: U1A promoter; pA: polyA; NLS: nuclear localization signal; HA: HA tag. (**B**) Protein expression levels of Nsp2Cas9, Nsp2-SmuCas9, and SpCas9 were analyzed by western blot. HEK293T cells without Cas9 transfection were used as a negative control (NC). (**C**) The editing efficiencies of Nsp2Cas9, Nsp2-SmuCas9, and SpCas9 varied depending on the target sites. The PAM sequences (NGGNCC) were shown below. Data represent mean ± SD for

*Figure 5 continued on next page*

*Figure 5 continued*

n=3 biologically independent experiments. (**D**) Quantification of the indel efficiencies for Nsp2Cas9, Nsp2-SmuCas9, and SpCas9. Each dot represents an average efficiency for an individual locus. Data represent mean ± SD for n=3 biologically independent experiments. p values were determined using a two-sided Student's *t* test. p=0.3883, p=0.7316, p=0.0741 (p>0.05), ns stands for not significant. *p=0.0247, *p=0.0144, (0.01<p < 0.05), * stands for significant. **p=0.0058 (p<0.01), ** stands for significant.

The online version of this article includes the following source data for figure 5:

**Source data 1.** Source data for *Figure 5B*.

**Source data 2.** Source data for the *Figure 5C and D*.

Lipofectamine 3000 (Life Technologies) according to the manufacturer's instructions. For transient transfection, a total of 300 ng Cas9-expressing plasmid and 200 ng sgRNA plasmid were co-transfected into a 48-well plate. For Cas9 PAM sequence screening, $1.2 \times 10^7$ HEK293T cells were transfected with 10 µg of Cas9 plasmid and 5 µg of sgRNA plasmid in 10 cm dishes.

## Flow cytometry analysis

Transfected library cells with a certain percentage of GFP-positive cells were collected by centrifugation at 1000 rpm for 5 min and resuspended in PBS. Then, GFP-positive cells were collected by flow cytometry and cultured in six-well plates. Five days after culture, we extracted the genome and built deep sequencing libraries.

## PAM sequence analysis

Twenty-base-pair sequences (AAGCCTTGTTTGCCACCATG/GTGAGCAAGG GCGAGGAGCT) flanking the target sequence (GAGAGTAGAGGCGGCCACGACCTGNNNNNNNN) were used to fix the target sequences. CTG and GTGAGCAAGGGCG AGGAGCT were used to fix 8 bp random sequences. Target sequences with in-frame mutations were used for PAM analysis. The 8 bp random sequence was extracted and visualized by WebLogo (*Crooks et al., 2004*) and a PAM wheel chart to identify PAMs (*Leenay et al., 2016*). The median coverage of every individual PAM variant and the number of unique PAM sequences are listed in *Figure 2—source data 1*.

## Genome editing and deep sequencing analysis of indels for endogenous sites

Cells were seeded into 48-well plates one day prior to transfection and transfected at 70–80% confluency using Lipofectamine 2000 (Life Technologies) following the manufacturer's recommended protocol. For genome editing, $10^5$ cells were transfected with a total of 300 ng of Cas9 plasmid and 200 ng of sgRNA plasmid in 48-well plates. Five days after transfection, the cells were harvested, and genomic DNA was extracted in QuickExtract DNA Extraction Solution (Epicenter). To measure indel frequencies, the target sites were amplified by two rounds of nested PCR to add the Illumina adaptor sequence. The PCR products (~400 bp in length) were gel-extracted by a QIAquick Gel Extraction Kit (QIAGEN) for deep sequencing.

## Western blot analysis

HEK293T cells were seeded into 24-well plates. The next day, the Cas9-expressing plasmid (800 ng) was transfected into cells using Lipofectamine 2000 (Invitrogen). Three days after transfection, cells were collected and resuspended in cell lysis buffer for Western blotting and IP (Beyotime) supplemented with 1 mM phenylmethanesulfonyl fluoride (PMSF) (Beyotime). Cell lysates were then centrifuged at 12,000 rpm for 20 min at 4 °C, and the supernatants were collected and mixed with 5 x loading buffer followed by boiling at 95 °C for 10 min. Equal amounts of protein samples were subjected to SDS-PAGE, followed by transfer to polyvinylidene difluoride (PVDF) membranes (Merck, Darmstadt, Germany). The PVDF membranes were blocked with 5% BSA in TBST for 1 hr at room temperature and then incubated with the anti-HA antibody (1:1000; Abcam) and anti-GAPDH antibody (1:1000; Cell Signaling) at 4 °C overnight.

The membrane was washed three times in TBS-T for 5 min each time. The membranes were incubated in the secondary goat anti-rabbit antibody (1:10,000; Abcam) for 1 hr at room temperature. The membranes were then washed with TBST buffer three times and imaged.

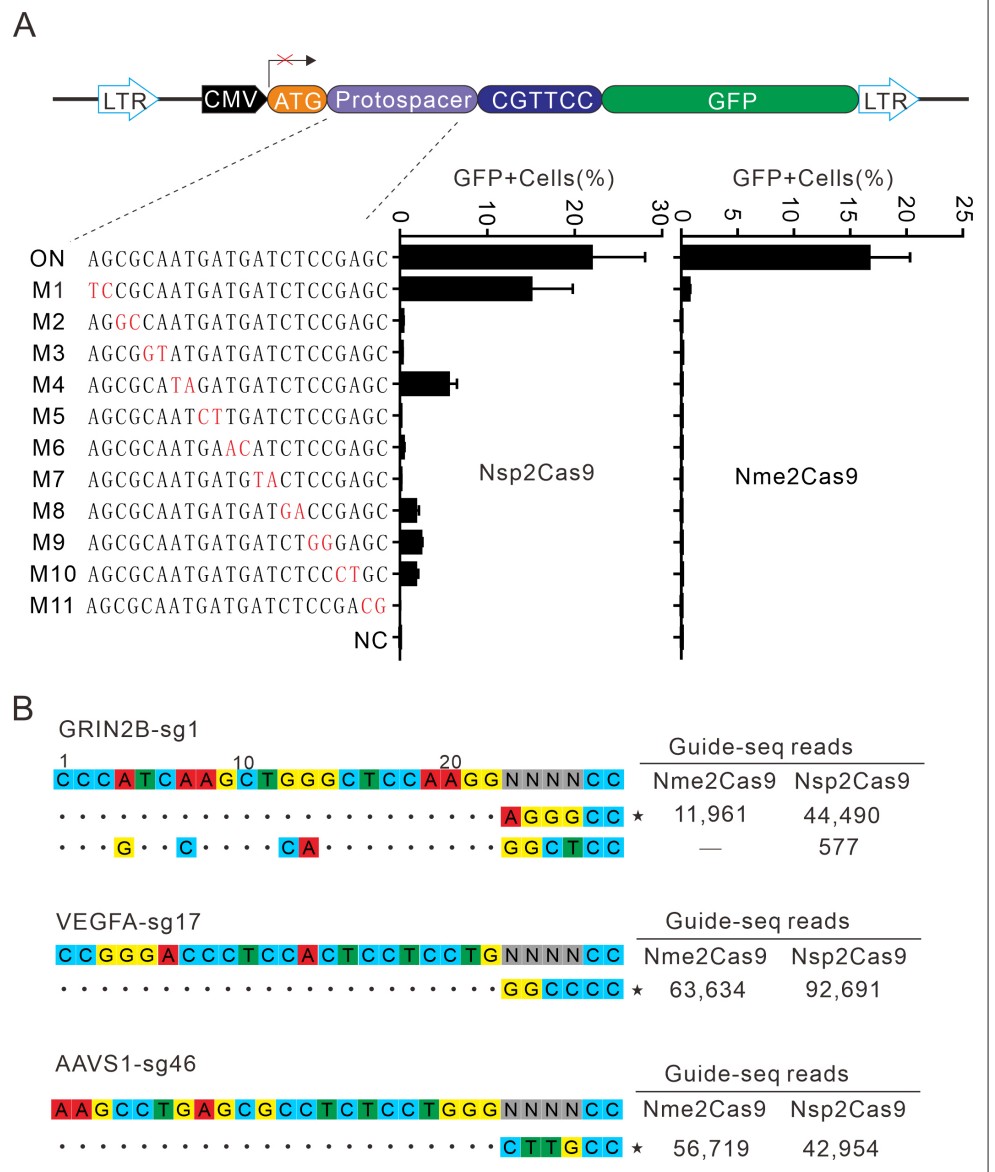

**Figure 6.** Analysis of Nsp2Cas9 specificity. (**A**) Analysis of Nsp2Cas9 and Nme2Cas9 specificity with a GFP-activation assay. A panel of sgRNAs with dinucleotide mutations (red) is shown below. The editing efficiencies reflected by ratio of GFP-positive cells are shown. Data represent mean ± SD for n=3 biologically independent experiments. (**B**) GUIDE-seq was performed to analyze the genome-wide off-target effects of Nsp2Cas9 and Nme2Cas9. On-target (indicated by stars) and off-target sequences are shown on the left. Read numbers are shown on the right. Mismatches compared to the on-target site are shown and highlighted in color.

The online version of this article includes the following source data for figure 6:

**Source data 1.** Source data for the *Figure 6A*.

## Test of Cas9 specificity

To test the specificity of Nsp2Cas9 and Nsp2-SmuCas9, we generated a GFP reporter cell line with a fixed PAM (5'-CGTTCC-3'). HEK293T cells were seeded into 48-well plates and transfected with 300 ng of Cas9 plasmids and 200 ng of sgRNA plasmids by using Lipofectamine 2000. Five days after transfection, the GFP-positive cells were digested and centrifuged at 1000 rpm for 3 min, and the cells were resuspended in phosphate-buffered saline (PBS). Finally, the cells were analyzed on a Calibur instrument (BD). The data were analyzed using the FlowJo software.

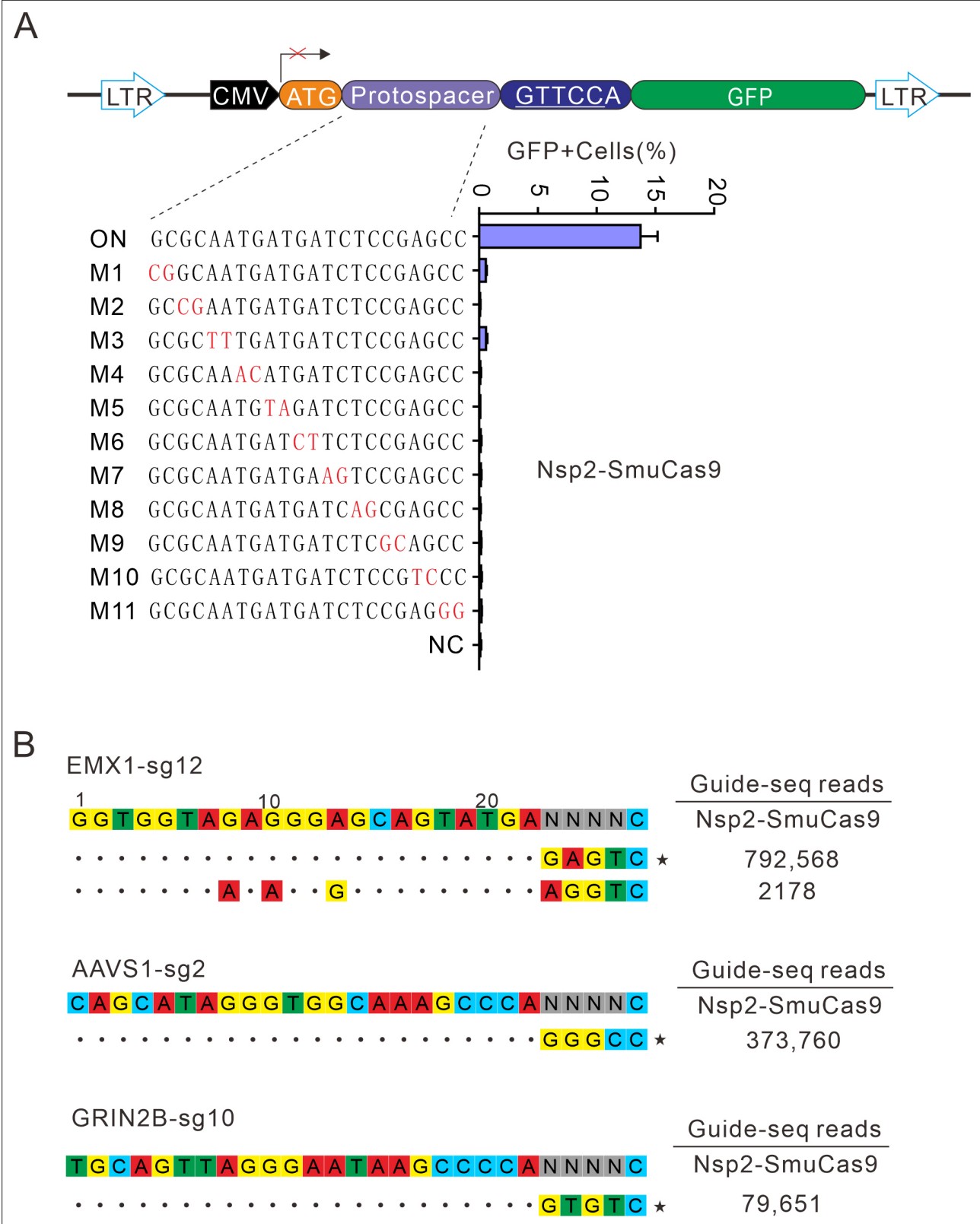

**Figure 7.** Analysis of Nsp2-SmuCas9 specificity. (**A**) Analysis of Nsp2-SmuCas9 specificity with a GFP-activation assay. A panel of sgRNAs with dinucleotide mutations (red) is shown below. The editing efficiencies reflected by ratio of GFP-positive cells are shown. Data represent mean ± SD for n=3 biologically independent experiments. (**B**) GUIDE-seq was performed to analyze the genome-wide off-target effects of Nsp2-SmuCas9. On-target

*Figure 7 continued on next page*

*Figure 7 continued*

(indicated by stars) and off-target sequences are shown on the left. Read numbers are shown on the right. Mismatches compared to the on-target site are shown and highlighted in color.

The online version of this article includes the following source data for figure 7:

**Source data 1.** Source data for the *Figure 7A*.

## GUIDE-seq

We performed a GUIDE-seq experiment with some modifications to the original protocol, as described (*Tsai et al., 2015*). On the day of the experiment, $2x10^5$ HEK293T cells per target site were harvested and washed in PBS and transfected with 500 ng of Cas9 plasmid, 500 ng of sgRNA plasmid and 100 pmol annealed GUIDE-seq oligonucleotides through the Neon Transfection System. The electroporation voltage, width, and number of pulses were 1150 V, 20ms, and 2 pulses, respectively. Genomic DNA was extracted with the DNeasy Blood and Tissue kit (QIAGEN) 6 days after electroporation according to the manufacturer's protocol. The genome library was prepared and subjected to deep sequencing.

## Statistical analysis

All data are presented as the mean ± SD. Statistical analysis was conducted using GraphPad Prism 7. Student's t test or one-way analysis of variance (ANOVA) was used to determine statistical significance between two or more than two groups, respectively. A value of $p<0.05$ was considered to be statistically significant (*$p<0.05$, **$p<0.01$, ***$p<0.001$). Raw data from all reported summary statistics were listed in *Supplementary file 5*.

## Acknowledgements

We thank supports from mRNA innovation and translation center, Shanghai. This work was supported by grants from the National Key Research and Development Program of China (2021YFA0910602, 2021YFC2701103); the National Natural Science Foundation of China (82070258, 81870199); Open Research Fund of State Key Laboratory of Genetic Engineering, Fudan University (No. SKLGE-2104), Science and Technology Research Program of Shanghai (19DZ2282100); the Natural Science Fund of Shanghai Science and Technology Commission (19ZR1406300).

## Additional information

### Competing interests

Siqi Gao, Yongming Wang: author on patent application number 202110878452X, "Cas9 protein, gene editing system containing Cas9 protein and application"; this patent relates to the technical field of gene editing, and specifically designs a CRISPR/Cas9 gene editing system and its application. The other authors declare that no competing interests exist.

### Funding

| Funder | Grant reference number | Author |
|---|---|---|
| National Key Research and Development Program of China | 2021YFA0910602 | Yongming Wang |
| National Natural Science Foundation of China | 82070258 | Yongming Wang |
| Fudan University | No. SKLGE-2104 | Yongming Wang |
| Shanghai Association for Science and Technology | 19DZ2282100 | Yongming Wang |
| Shanghai Science and Technology Committee | 19ZR1406300 | Yongming Wang |

| Funder | Grant reference number | Author |
| --- | --- | --- |
| National Key Research and Development Program of China | 2021YFC2701103 | Yongming Wang |
| National Natural Science Foundation of China | 81870199 | Yongming Wang |

The funders had no role in study design, data collection and interpretation, or the decision to submit the work for publication.

## Author contributions

Jingjing Wei, Data curation, Methodology, Writing - original draft; Linghui Hou, Siqi Gao, Methodology; Jingtong Liu, Tao Qi, Data curation; Ziwen Wang, Song Gao, Investigation; Shuna Sun, Resources; Yongming Wang, Resources, Formal analysis, Supervision, Project administration, Writing – review and editing

## Author ORCIDs

Ziwen Wang http://orcid.org/0000-0002-1678-7624
Song Gao http://orcid.org/0000-0001-7427-6681
Yongming Wang http://orcid.org/0000-0001-8269-5296

## Decision letter and Author response

Decision letter https://doi.org/10.7554/eLife.77825.sa1
Author response https://doi.org/10.7554/eLife.77825.sa2

---

## Additional files

### Supplementary files

- Supplementary file 1. The Cas9 ID and human codon–optimized Cas9 gene. The file contains the Cas9 ID, tracrRNA, and amino acid sequences of Nme1Cas9 orthologs used in this study. The human codon-optimized Cas9 genes were synthesized. The primers used for the chimera's construction were also listed in this file.

- Supplementary file 2. The single-guide RNA sequence. The single-guide RNA sequence for Nme1Cas9 orthologs and chimeras used in this study.

- Supplementary file 3. Primers used in this study. A list of oligonucleotide pairs and primers used for deep sequencing.

- Supplementary file 4. Target sites used in this study. A list of the endogenous target sites of human and mouse and their downstream PAM. PAM, protospacer adjacent motif.

- Supplementary file 5. Underlying values for all reported summary statistics. Raw data from all reported summary statistics.

- Transparent reporting form

### Data availability

All data generated or analysed during this study are included in the manuscript and supporting file.

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
