## [Editor Report]

This work is relevant to all who are interested in genome editing. The versatile Cas9 nuclease has enabled creative genome editing applications, yet the targetable sequence space is limited by the PAM specificity of the Cas9 RNP. This manuscript expands the Cas9 toolbox by defining the PAM specificity and genome editing activity of a large group of smaller-sized type II-C Cas9s. The results also contribute to our understanding of the diversity of Cas enzymes and show that there is a significant potential in mining for non-trivial genome editing tools amongst highly similar Cas orthologs.

---

## [Decision Letter]

**Decision letter after peer review:**

Thank you for submitting your article "Closely related type II-C Cas9 orthologs recognize diverse PAMs" for consideration by *eLife*. Your article has been reviewed by 3 peer reviewers, and the evaluation has been overseen by a Reviewing Editor and Kevin Struhl as the Senior Editor. The following individual involved in review of your submission has agreed to reveal their identity: Konstantin Severinov (Reviewer #3).

Essential revisions:

1) As suggested in individual reviewer critiques (provided below), please provide clarification on the editing efficiency of the II-C Cas9s described here. If possible, a comparison to a commonly used standard (e.g,. SpCas9) would be a worthwhile addition.

2) Please revise the manuscript introduction and Discussion sections as outlined by individual critiques to include additional relevant background information and provide important caveats or points of clarification.

3) Please include sequences for the various constructs used here, and address other comments specified in individual reviewer critiques.

*Reviewer #1 (Recommendations for the authors):*

Figure 2 and supplement: The main Figure 2 and figure supplement 1 are almost identical. There are some subtle differences in the logos, but it's not clear to me why both figures are included and why there are differences. It seems like maybe they are shifted over in some cases, and it would be helpful to decide if we really need both images and precisely clarify the difference if both remain.

Line 46: it is not a problem for analyzed orthologs to be "phylogenetically distinct".

Line 46: "The existing evidence... " has no citation. The evidence or logic behind this claim is not clear to the reader.

Line 99,132 (and throughout): Use a prime symbol, not an apostrophe.

Line 198: This title indicates that Nsp2-SmuCas9 is being assessed for specificity and instead this section is initially about Nme2Cas9. The title is confusing in light of the writing in the section.

Line 222: It is not clear why they remain to be improved. They seem like they work to a large extent. Ending with this sentence detracts from the excitement and positive outcomes in the work. Further, it probably belongs in the discussion.

*Reviewer #2 (Recommendations for the authors):*

I have the following points for the authors to address:

1. Overall, the editing efficiency of II-C Cas9s were in the 10-30% range. Was it due to the assay setup, or did it reflect the intrinsic property of II-C Cas9s? Have the authors made a side-by-side comparison with SpCas9 under the same assay conditions?

2. While Nsp2Cas9 chimera displayed an impressively degenerate PAM specificity, its fidelity was not as impressive as the benchmark (Nme2Cas9). Have the authors made any effort to generate Nme2Cas9-Smu? Expanding the PAM code of an existing hi-fi Cas9 is still quite significant.

*Reviewer #3 (Recommendations for the authors):*

1. In the Method section "PAM sequence analysis" information about median coverage of every individual PAM variant shall be provided. How many unique PAM sequences were used to build PAM logo to support the obtained results?

2. Perhaps low DNA cleavage activity of some proteins is due to low level of their production. Was the synthesis of each of the Cas9 orthologs tested by Western Blot prior to PAM screening in human cells?

3. Figure 3—figure supplement 3A shows that Cas9 proteins contained four NLS signals. It is not clear if these are different signal and why so many NLS were used. I would recommend to include maps of all plasmids used in the study through Benchling or other tools.

4. The authors use 22nt spacer segment sgRNA in their work. The choice of this length can be confusing for people who work with SpCas9. I would recommend to emphasize the importance the sgRNAs of this length with the enzymes studied.

5. I would recommend to extend the Introduction section and add information on other II-C Cas9 orthologs characterized to date.

---

## [Author Response]

Essential revisions:1) As suggested in individual reviewer critiques (provided below), please provide clarification on the editing efficiency of the II-C Cas9s described here. If possible, a comparison to a commonly used standard (e.g,. SpCas9) would be a worthwhile addition.

We sincerely thank the editor for the constructive comments on our manuscript. Following editor’s suggestions, we compared the editing efficiency of Nsp2Cas9, Nsp2-SmuCas9, SpCas9, SpCas9-NG, and SpCas9-RY side-by-side. Overall, the editing efficiency was low this time probably due to low transfection efficiency. The results revealed that SpCas9 was the most active enzyme. Nsp2Cas9, SpCas9-NG, and SpCas9-RY displayed similar activity. Nsp2-SmuCas9 displayed lower activities than other Cas9 variants (Figure 5C).

2) Please revise the manuscript introduction and Discussion sections as outlined by individual critiques to include additional relevant background information and provide important caveats or points of clarification.

We have added the additional relevant background information and provide important caveats or points of clarification in the “introduction” and “discussion” sections and highlighted in yellow according the views of the reviewers.

3) Please include sequences for the various constructs used here, and address other comments specified in individual reviewer critiques.

The sequences for the various constructs used here are listed in Supplementary file 1. The maps of all plasmids used in the study are packaged in Figure 1-source data 1. We have attached a point-to-point response to each of the reviewers’ comments and hope that the revised manuscript is now suitable for publication in *eLife*.

Reviewer #1 (Recommendations for the authors):Figure 2 and supplement: The main Figure 2 and figure supplement 1 are almost identical. There are some subtle differences in the logos, but it's not clear to me why both figures are included and why there are differences. It seems like maybe they are shifted over in some cases, and it would be helpful to decide if we really need both images and precisely clarify the difference if both remain.

Thank you for your suggestions. For the sake of simplicity, we deleted Figure 2—figure supplement 1 in the revised manuscript.

Line 46: it is not a problem for analyzed orthologs to be "phylogenetically distinct".

Thank you for your suggestions. In the revised manuscript, we deleted this sentence.

Line 46: "The existing evidence... " has no citation. The evidence or logic behind this claim is not clear to the reader.

Thank you for your suggestions. In the revised manuscript, we stated “Several studies have revealed that even phylogenetically closely related Cas9 orthologs may recognize distinct PAMs”. The citation has been added.

Line 99,132 (and throughout): Use a prime symbol, not an apostrophe.

The prime symbol has been used in the revised manuscript.

Line 198: This title indicates that Nsp2-SmuCas9 is being assessed for specificity and instead this section is initially about Nme2Cas9. The title is confusing in light of the writing in the section.

Thank you for your suggestions. In the revised manuscript, we stated “Analysis of Nsp2Cas9 specificity”.

Line 222: It is not clear why they remain to be improved. They seem like they work to a large extent. Ending with this sentence detracts from the excitement and positive outcomes in the work. Further, it probably belongs in the discussion.

Thank you for your suggestions. In the revised manuscript, we stated “Altogether, these results demonstrated that Nsp2Cas9 and Nsp2-SmuCas9 offered new platforms for genome editing.”.

Reviewer #2 (Recommendations for the authors):I have the following points for the authors to address:1. Overall, the editing efficiency of II-C Cas9s were in the 10-30% range. Was it due to the assay setup, or did it reflect the intrinsic property of II-C Cas9s? Have the authors made a side-by-side comparison with SpCas9 under the same assay conditions?

We sincerely thank the reviewer for the constructive comments on our manuscript. Following reviewer’s suggestions, we compared the editing efficiency of Nsp2Cas9, Nsp2-SmuCas9, SpCas9, SpCas9-NG, and SpCas9-RY side-by-side. Overall, the editing efficiencies were low this time probably due to low transfection efficiency. The results revealed that SpCas9 was the most active enzyme. Nsp2Cas9, SpCas9-NG, and SpCas9-RY displayed similar activity. Nsp2-SmuCas9 displayed lower activities than other Cas9 variants (Figure 5C).

Gong et al. have shown that DNA Unwinding by Cas9 is the rate-limiting step in target cleavage [5]. Ma et al. have shown that type II-C Cas9s display lower DNA unwinding activities [6]. These results indicate that II-C Cas9s generally display lower activity than II-A Cas9s. Consistently, our lab screened dozens of II-C Cas9s, but failed to identify a Cas9 with activity comparable to SpCas9. In contrast, our lab screened dozens of II-A Cas9s and identified three Cas9s, including SlugCas9 [7], with activity comparable to SpCas9.

2. While Nsp2Cas9 chimera displayed an impressively degenerate PAM specificity, its fidelity was not as impressive as the benchmark (Nme2Cas9). Have the authors made any effort to generate Nme2Cas9-Smu? Expanding the PAM code of an existing hi-fi Cas9 is still quite significant.

Thank you for your suggestions. In the revised manuscript, we generated a chimeric Nme2-SmuCas9. However, Nme2-SmuCas9 activity was very low. Of 12 targets tested, only two targets could be edited (Figure 4—figure supplement 1). We further generated chimeric Nme2-NarCas9 and Nsp2-NarCas9, but they did not work. Chimeric Cas9s generally display reduced activity. We previously generated several SaCas9-based chimeras, but only Sa-SlugCas9 displayed robust activity [7].

Reviewer #3 (Recommendations for the authors):1. In the Method section "PAM sequence analysis" information about median coverage of every individual PAM variant shall be provided. How many unique PAM sequences were used to build PAM logo to support the obtained results?

We sincerely thank the reviewer for the constructive comments on our manuscript. Following reviewer’s suggestions, we provided the sequences used for PAM analysis in Figure 2-Source Data 1. The number of unique PAM sequences varied from 518 to 2,193.

2. Perhaps low DNA cleavage activity of some proteins is due to low level of their production. Was the synthesis of each of the Cas9 orthologs tested by Western Blot prior to PAM screening in human cells?

Thank you for your suggestions. Following reviewer’s suggestions, we performed Western Blot to measure the expression of each Cas9 protein, and results revealed that expression levels of these Cas9s were comparable (Figure 1—figure supplement 5).

3. Figure 3—figure supplement 3A shows that Cas9 proteins contained four NLS signals. It is not clear if these are different signal and why so many NLS were used. I would recommend to include maps of all plasmids used in the study through Benchling or other tools.

Thank you for your suggestions. The Nme2Cas9_AAV vector [8] generated by Sontheimer group contains four NLS signals. More NLS signals may enhance the nuclear entry of the proteins and increase the editing activity. In this study, the Nme2Cas9_AAV backbone was used for our Cas9 expression. The maps of all plasmids used in the study are packaged in Figure 1-source data 1.

4. The authors use 22nt spacer segment sgRNA in their work. The choice of this length can be confusing for people who work with SpCas9. I would recommend to emphasize the importance the sgRNAs of this length with the enzymes studied.

Many thanks for the reviewer’s comments. The length of the sgRNA can influence editing efficiencies. Amrani et al. had tested the sgRNA length for Nme1Cas9, and observed comparable activities with 21-24 nt sgRNAs. In this study, we tested the sgRNA length from 18 to 26 nt in HEK293T cells, and the results showed that 22-26 nt sgRNA activities were comparable (Figure 3—figure supplement 1). We used 22 nt in our study.

5. I would recommend to extend the Introduction section and add information on other II-C Cas9 orthologs characterized to date.

Many thanks for the reviewer’s suggestions. Other II-C Cas9s have been added in the Introduction “Several type II-C Cas9s, including NmeCas9 [9], Nme2Cas9 [8], CjCas9 [10], GeoCas9 [11], BlatCas9 [12], and PpCas9 [13] have been developed for genome editing.

References

1. Kleinstiver BP, Prew MS, Tsai SQ, Nguyen NT, Topkar VV, Zheng Z, et al. Broadening the targeting range of *Staphylococcus aureus* CRISPR-Cas9 by modifying PAM recognition. Nature biotechnology. 2015;33(12):1293-8. doi: 10.1038/nbt.3404. PubMed PMID: 26524662; PubMed Central PMCID: PMC4689141.

2. Walton RT, Christie KA, Whittaker MN, Kleinstiver BP. Unconstrained genome targeting with near-PAMless engineered CRISPR-Cas9 variants. Science. 2020;368(6488):290-6. Epub 2020/03/29. doi: 10.1126/science.aba8853. PubMed PMID: 32217751; PubMed Central PMCID: PMCPMC7297043.

3. Ran FA, Cong L, Yan WX, Scott DA, Gootenberg JS, Kriz AJ, et al. in vivo genome editing using *Staphylococcus aureus* Cas9. Nature. 2015;520(7546):186-91. doi: 10.1038/nature14299. PubMed PMID: 25830891; PubMed Central PMCID: PMC4393360.

4. Wang S, Mao HL, Hou LH, Hu ZY, Wang Y, Qi T, et al. Compact SchCas9 Recognizes the Simple NNGR PAM. Adv Sci. 2021. doi: Artn 2104789

10.1002/Advs.202104789. PubMed PMID: WOS:000727233600001.

5. Gong S, Yu HH, Johnson KA, Taylor DW. DNA Unwinding Is the Primary Determinant of CRISPR-Cas9 Activity. Cell Rep. 2018;22(2):359-71. Epub 2018/01/11. doi: 10.1016/j.celrep.2017.12.041. PubMed PMID: 29320733.

6. Ma E, Harrington LB, O'Connell MR, Zhou K, Doudna JA. Single-Stranded DNA Cleavage by Divergent CRISPR-Cas9 Enzymes. Mol Cell. 2015;60(3):398-407. doi: 10.1016/j.molcel.2015.10.030. PubMed PMID: 26545076; PubMed Central PMCID: PMC4636735.

7. Hu Z, Zhang C, Wang S, Gao S, Wei J, Li M, et al. Discovery and engineering of small SlugCas9 with broad targeting range and high specificity and activity. Nucleic Acids Res. 2021;49(7):4008-19. doi: 10.1093/nar/gkab148. PubMed PMID: 33721016; PubMed Central PMCID: PMC8053104.

8. Edraki A, Mir A, Ibraheim R, Gainetdinov I, Yoon Y, Song CQ, et al. A Compact, High-Accuracy Cas9 with a Dinucleotide PAM for in vivo Genome Editing. Mol Cell. 2019;73(4):714-26 e4. doi: 10.1016/j.molcel.2018.12.003. PubMed PMID: 30581144; PubMed Central PMCID: PMC6386616.

9. Hou Z, Zhang Y, Propson NE, Howden SE, Chu LF, Sontheimer EJ, et al. Efficient genome engineering in human pluripotent stem cells using Cas9 from Neisseria meningitidis. Proceedings of the National Academy of Sciences of the United States of America. 2013;110(39):15644-9. doi: 10.1073/pnas.1313587110. PubMed PMID: 23940360; PubMed Central PMCID: PMC3785731.

10. Kim E, Koo T, Park SW, Kim D, Kim K, Cho HY, et al. in vivo genome editing with a small Cas9 orthologue derived from Campylobacter jejuni. Nature communications. 2017;8:14500. doi: 10.1038/ncomms14500. PubMed PMID: 28220790; PubMed Central PMCID: PMC5473640.

11. Harrington LB, Paez-Espino D, Staahl BT, Chen JS, Ma E, Kyrpides NC, et al. A thermostable Cas9 with increased lifetime in human plasma. Nature communications. 2017;8(1):1424. Epub 2017/11/12. doi: 10.1038/s41467-017-01408-4. PubMed PMID: 29127284; PubMed Central PMCID: PMCPMC5681539.

12. Gao N, Zhang C, Hu Z, Li M, Wei J, Wang Y, et al. Characterization of Brevibacillus laterosporus Cas9 (BlatCas9) for Mammalian Genome Editing. Frontiers in cell and developmental biology. 2020;8:583164. doi: 10.3389/fcell.2020.583164. PubMed PMID: 33195228; PubMed Central PMCID: PMC7604293.

13. Fedorova I, Vasileva A, Selkova P, Abramova M, Arseniev A, Pobegalov G, et al. PpCas9 from Pasteurella pneumotropica – a compact Type II-C Cas9 ortholog active in human cells. Nucleic Acids Res. 2020;48(21):12297-309. Epub 2020/11/06. doi: 10.1093/nar/gkaa998. PubMed PMID: 33152077; PubMed Central PMCID: PMCPMC7708072.